# PURSUIT POLICIES IN DYNAMIC ENVIRONMENTS

## ABSTRACT

Cooperative pursuit is a popular multi-agent reinforcement learning (MARL) game where a team of predators target prey while avoiding obstacles. Previous literature has largely considered the impact of different predator, prey abilities on learning. Here, we investigate the impact of dynamic environments on learning predator pursuit policies from partial observations with deep Q-learning. Interestingly, we find predators are able to learn cooperative pursuit strategies that leverage moving obstacles.

## 1 INTRODUCTION

Multi-agent reinforcement learning (MARL) is a sub-field of reinforcement learning (RL) that focuses on multiple agents learning simultaneously in a shared environment. Within MARL, there many popular environments that are used as a test-bed for reinforcement learning research. Here, our investigation focuses on the cooperative pursuit game, played in the grid-world with two teams, predators and prey, and the ability of predators to learn with different types of obstacles.

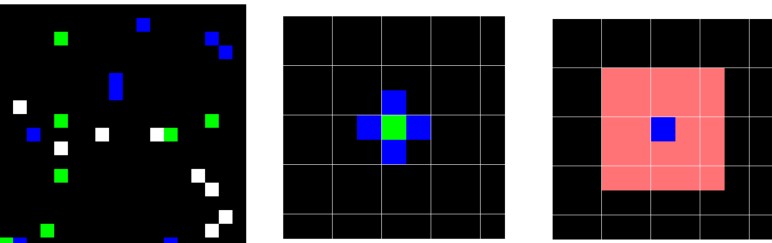

(a) An example of a discrete pursuit environment. (b) Successful capture by four predators (blue). (c) Local vision (red) of a predator (blue).

Figure 1: Examples of environments rendered from experiments. Obstacles in white, predators in blue, and prey in green. Example predator vision shown with red.

There has been extensive work within MARL that has motivated this work. Cooperative pursuit has been a popular multi-agent game to investigate algorithmic performance [Parsons (1978), Chung et al. (1988), Lin (1992), Omidshafiei et al. (2017), Dibangoye & Buffet (2018), Yu et al. (2020), Vidal et al. (2002), Hollinger et al. (2010), Kehagias et al. (2009), Lowe et al. (2020), Lanctot et al. (2017)]. Moreover, MARL research has been augmented by general advances in RL [van Hasselt et al. (2015), Wang et al. (2016), Sunehag et al. (2017), Yu et al. (2015), Travnik et al. (2018), Schaul et al. (2016), Mnih et al. (2013), Svetlik et al. (2017), OpenAI et al. (2019), Jaderberg et al. (2019), Sukhbaatar et al. (2018), Leibo et al. (2019), Gronauer & Diepold (2022), Pathak et al. (2017), Bertsekas (2012)]. Further, the investigation of dynamic environments is inspired by previous experimental success in learning policies [Gottlieb & Shima (2015), Baker et al. (2020), Wang et al. (2022)]; namely, this work is inspired by the learned behavior seen in the MARL game "Hide-and-Seek" [Baker et al. (2020)] where agents have non-local, line-of-sight vision. Experiments are programmed independently for performance with *Tensorflow* (Abadi et al. (2015)).

Table 1: Summary of training method used in experiments

| Parameter type | Parameter value |
| --- | --- |
| Architecture | Deep Q-Network (DQN) Input (5x5), Hidden Layer (16), Output (4) |
| Episodes | 100 |
| Rounds per episode | 20 |
| Learning rate ($\alpha$) | 0.0001 |
| Epsilon ($\epsilon$) | 0.1 |
| Discount factor ($\gamma$) | 0.8 |

## 2 METHODS AND RESULTS

Learning of cooperative pursuit policies takes place over numerous, successive games that are simulated in our programmed environment [1]. At the start of each game, each agent is given identical networks to make decisions with. Throughout the game, each agent maintains a memory of its experiences using its two-hop vision and executes an $\epsilon$-greedy algorithm to aid exploration. Games are played for 20 steps, then each agent's game experience replay is sampled to train the base network. The network is trained with a custom reward function that gives a reward of +10 for each agent capturing a prey, +1 for each each agent within two-hops of a prey, and -1 to all other actions. After training, agents memories are cleared and another game begins. Unlike predators, prey execute a random walk to an unoccupied position.

Our experiments' performance are measured by the collective reward of predators each game round, varying the dynamics of obstacles: static, periodic, and random. In the environment, with algorithm parameters, in Table 1. In each setting, a team of five predators move in a 15x15 grid-world environment where there are five obstacles and five prey. Predators, prey and obstacles are spawn in random positions to start. In the static case, obstacles do not move. In the periodic case, obstacles move periodically along the vertical direction. In the random case, obstacles move in a random walk. Our results are presented in Figure 2

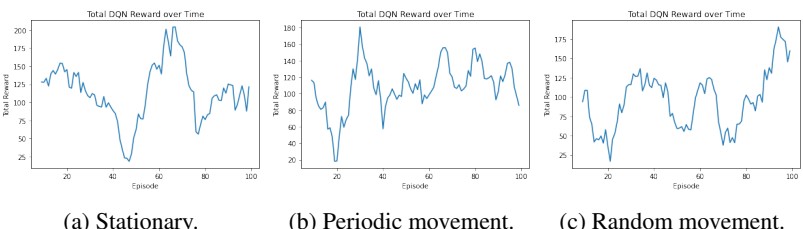

| (a) Stationary. | (b) Periodic movement. | (c) Random movement. |
| --- | --- | --- |

Figure 2: Reward yielded by games played with different obstacle movements. Rolling reward averaged over previous 10 games plotted.

Cooperative pursuit policies learned in dynamic environments preform as well, or better, than stationary counterparts, suggesting that observations in a dynamic environment can aid in learning.

## 3 CONCLUSION AND FUTURE WORK

Here, we investigate the impact of dynamic environments on learning predator pursuit policies from partial observations with deep Q-learning. Our preliminary investigation shows that policies learned in dynamic environments can out preform the ones learning in a static environment. This work can be expanded upon in a number of directions, including, but not limited to, testing algorithmic performance with prey that have the ability to learn, leveraging local clustering to aid pursuit, and investigating different deep learning architectures.

---

[1] Available at Github: Removed for review.

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

## 4 URM STATEMENT

Acknowledgement of meeting the URM criteria.

