# OpenReview forum: "Pursuit Policies in Dynamic Environments"
_ICLR.cc/2023/TinyPapers — Submitted to Tiny Papers @ ICLR 2023_

### Official Review · Reviewer_8Fa6 · 2023-03-19

**Confidence:** 2

**Summary Of Contributions:**

A paper that explores the usefulness of learning in dynamic environments over static environments. The paper show that when the environment's elements (such as obstacles) are randomized, agents achieve higher rewards.

**Rating:**

Clear, Correct, and Reproducible (CCR): a submission which meets the reviewing criteria

**Strengths And Weaknesses:**

STRENGTHS

- The problem is well motivated and the paper well written and easy to follow. The paper explains very well that Cooperative Pursuit is a well-known benchmark in MARL and in the multi-agent community in general.

- The problem if generalization in MARL is vastly unexplored and literature on the topic is very well received. This paper has great potential as it tackles a challenging topic.

- The experiments are well designed and the environments are well thought. There are several types of difficulties and different dynamics considered.

- Results support the claimed conclusion.

WEAKNESSES
- It is not clear the cooperative algorithm used. Whether we are in a MARL problem modeled as "independent learners" or what was the rationale behind training individual DQNs with individual rewards. It is better to use a cooperative algorithm for cooperative MARL problems.

- The results shown high variance on the returns, this is very likely a result of using DQN as it is known to present this oscillatory behavior.





**Suggested Changes:**

1. The article is very well written and easy to follow, which is a great merit per se. I would suggest some stylistic improvements to make the writing stronger. I would like to focus on best practices to reference other people's work. First, there are too many references for simple ideas. Usually, if you want to say that there is an area of active research that has brought a lot of attention, maybe reference a Survey paper on the topic, or  those kind of papers named "open challenges in .....". By referencing "survey papers" or "challenges papers" you save space and don't have to use 15 references for your idea. Or at least, consider removing some of the oldest references like Bertsekas 2012, Mnih 2013 (which is not about MARL) and others. Make your references short and focused, only use the ones necessary.

In the space that you save, explain to the reader that these experiments have not been done before (I assume) and thus your paper fills a gap in the literature.

2. On the same line of the above, "Tensorflow" doesn't need to be referenced, it's a well-known library.

The above recommendations will free up some space and enhance the readability of your paper.

3. It is great that you plan to make your code public. This is often a strong point for a paper. Just for your information, for future submissions, there is a way to anonymize GitHub repos, there are several instructions available online.

4. For future reference, look at other papers how the parameters of the experiments are presented. Usually, what you have put on Table 1 goes to the appendix, leaving more room in the central part of the paper. Usually papers have a section called "Experiment setup" where they define the hyperparameters used and all the settings. On the main paper just show your results. Then on an appendix, you usually show more results, ablation studies, etc.

5. It is not clear from the paper if this is a MARL problem or a single-agent setup. From the wording, this is a MARL environment where agents have to cooperate to achieve a goal. However, the training and the reward structure seems to be following a single-agent implementation. I will explain further: in multi-agent settings, you can very well have one policy net per agent, like in this case, where each agent has its own DQN. This "independent learning" has its pros and cons, but it's a known alternative. However, it is hardly MARL as there is no cooperation between the agents. I would recommend to at least mention that you are following the "independent-learning" paradigm (you can read more here or use this article as a citation: https://www.frontiersin.org/articles/10.3389/frai.2022.805823/full) and explain your choices. There are currently open source implementation of other MARL algorithms that promote cooperation among agents, one of such it's "Multi-Agent PPO".

6. Explain better the role of the "base network". Maybe on the appendix if there is no more space left.

---

### Official Review · Reviewer_HjMx · 2023-03-24

**Confidence:** 4

**Summary Of Contributions:**

The paper experimentally studies the impact of a dynamic environment on the learning of predators in a cooperative pursuit game.

**Rating:**

Needs Clarification (NC): a submission which does not meet the reviewing criteria and needs clarification for its described problem or solution

**Strengths And Weaknesses:**

The impact of a dynamic environment on the learning of agents in cooperative pursuit is an interesting and well-motivated problem to study. However, the experimental results are not strong enough yet to draw any meaningful conclusion. In the simplest stationary setting, this is a small-scale multi-agent RL experiment without any change in the dynamics. I would expect the predators to learn very well over time. Surprisingly, the cumulative reward collected by the predators are just fluctuating without showing any real signs of learning. How much cumulative reward is collected by a completely random policy? The authors should consider checking their algorithm implementations. Figures 2 b and c are also very noisy plots. If this plot is from a single run of training, I suggest the authors performing a larger number of runs and plot the mean and std of the cumulative rewards.

**Suggested Changes:**

Please see the comments above.

---

### Meta-Review · Area_Chair_Vi3u · 2023-04-07

**Recommendation:** Invite to revise
**Confidence:** 4

**Metareview:**

The paper addresses the challenge of generalization in multi-agent reinforcement learning (MARL) by studying the impact of a dynamic environment on the learning of predators in a cooperative pursuit game. The experiments explore the usefulness of learning in dynamic environments compared to static environments and demonstrate that when the environment's elements, such as obstacles, are randomized, agents achieve higher rewards. There are some areas that could be improved, such as more clarity on the cooperative algorithm used and the role of the base network. The paper's significance lies in its contribution to understanding the usefulness of learning in dynamic environments, which could have important implications for real-world applications of MARL.

**Summary:**

The paper investigates the impact of a dynamic environment on the learning of predators in a cooperative pursuit game.

**Comments And Feedback To The Authors:**

I would like to commend you on your well-written paper and interesting findings on the impact of a dynamic environment on the learning of predators in a cooperative pursuit game. The problem you are addressing is well-motivated, and the experiments are well-designed.

However, I have some recommendations that I believe would improve the quality and clarity of your paper. Firstly, it would be helpful to provide more clarification on the cooperative algorithm used and whether the paper is addressing a single-agent or MARL problem. Additionally, the high variance of the results and the need for more clarity on the role of the base network are also areas that could be improved. You could also benefit from using fewer references for simple ideas and explaining better the gap in the literature that your work fills. I would also suggest considering the implications of your findings for real-world applications of MARL and performing more robustness analysis, such as sensitivity analysis to hyperparameters and the effect of different levels of randomness in the environment. Two reviewers' comments are also very valuable. Please try to answer Reviewer HjMx's concerns in the revised paper.

**Reason For Not Giving A Higher Recommendation:**

The reason for not giving a higher recommendation is due to the weaknesses and areas for improvement identified in the paper, such as the lack of clarity on the cooperative algorithm used and the need for more clarification on whether the paper is addressing a single-agent or MARL problem. The high variance in the results and the need for more clarity on the role of the base network are also areas that could be improved. Additionally, the paper could benefit from more concise referencing and better explanation of the gap in the literature that it fills.

**Reason For Not Giving A Lower Recommendation:**

N/A

---

### Decision · Program_Chairs · 2023-04-09

No revision received; not invited to archive